# Government Subsidy Policy and Online Selling Strategy in a Platform Supply Chain with Green R&D and DDM Activities

**Zongyu Mu *** , **Qiujie Li, Gengxin Dai, Ke Li, Guangming Zhang and Fan Zhang**

Department of Management Science and Engineering, School of Business, Qingdao University,
Qingdao 266071, China
* Correspondence: mzydragon@163.com; Tel.: +86-153-7689-2910

**Abstract:** Many governments actively subsidize the green activities of manufacturers and consumers to effectively realize the achievement of carbon emissions peak and carbon-neutral goals, while the development of a platform economy can effectively contribute to sustainable development. Therefore, we have modeled a platform supply chain using game theory, in which the manufacturer conducts green research and development (R&D) activities, the third-party platform conducts data-driven marketing (DDM) activities to promote green products, and all consumers have green preferences. The numerical example and empirical analysis methods are used to mine management insights. The government subsidizes the manufacturer's green R&D, the third-party platform's DDM, and the consumers' green consumption. The third-party platform provides an agency selling or reselling strategy to sell products. Our results show that: (1) the sensitivity coefficient of consumers to green R&D and DDM activities has positive impacts on all members' profits and on the green R&D level of products in the platform supply chain, with three kinds of government subsidy policies. (2) The levels of the three kinds of government subsidies mainly have an impact on all members' profits and on the green R&D level of products in the platform supply chain with an agency selling or reselling strategy; government subsidies to the manufacturer are more conducive to improving the green R&D level of products. (3) The levels of the three government subsidies and the unit service commissioning fee for selling products are the main factors affecting the preferred selling strategy of each member and the equilibrium of the selling strategy.

**Keywords:** platform supply chain; government subsidy; green R&D; data-driven marketing; online selling

## 1. Introduction

In 2021, global carbon emissions from manufacturing reached 36.3 billion tons, an increase of 6% over 2020 and the highest ever annual level. This level not only exacerbates global warming but also seriously damages the ecological environment [1,2]. These problems have seriously affected sustainable economic development around the world. Thus, more governments have begun to implement policies regarding the carbon emissions peak and reaching carbon neutral, to motivate manufacturers to carry out green production and promotion activities and consumers to buy green products [3–6]. For example, the German government encourages semiconductor chip manufacturers to produce environmentally friendly semiconductors by offering a subsidy of 14 billion euros. The Chinese government provides subsidies to the Taobao.com and JD.com platforms according to the energy-saving level of the products, to encourage these retailers to participate in energy conservation and emissions reduction. The Swedish government has introduced a green car subsidy policy of approximately USD 1500 for all consumers who buy green cars, which has effectively promoted the selling of environmentally friendly cars. These government subsidy policies are dedicated to addressing the devastating effects of global warming and environmental pollution in sustainable economic development and play an important role in improving the

operational efficiency of the green supply chain. At the same time, increasing consumers' awareness of green environmental protection also helps green research and development (R&D) and promotional activities in the supply chain. More and more manufacturers and retailers have realized that conducting green R&D activities can attract more market shares and gain more profits from consumers' green preferences, which is conducive to achieving win-win economic and ecological benefits [7–9].

With the development of Internet technology, the platform economy has injected new vitality into the traditional economy and upgraded the industrial structure to improve the operation efficiency of the supply chain. More and more manufacturers are acutely aware that an online business platform is the core of supply chain system construction and dictates the direction of optimization and upgrade; therefore, the platform supply chain arises [10,11]. There are two cooperative modes between the manufacturer and the third-party platform in agreeing to sell products in a platform supply chain system. One is the agency selling strategy, for example, TOPSPORTS. Sephora sell their products through JD.COM, which charges a service commission fee calculated on the selling price of the per unit product. The other is the reselling strategy; for example, Huawei, Coach, and Burberry sell their products to JD.COM at a wholesale price, and JD.COM sells them on to consumers at a marked-up price. Amazon.com not only provides a reselling channel to sell products but also runs an agency selling channel to sell products on behalf of manufacturers. An online business platform has the natural advantages of data collection that can be used to describe, predict, analyze, and guide consumers' purchase behavior according to the obtained green products data [12–15]; the integration of green concepts and the platform economy can take advantage of environmental protection and data-driven marketing (DDM) technology to help manufacturers to better balance economic development and ecological benefits [16]. In fact, Apple Inc. has already reduced the carbon footprint of its products by using renewable energy and has committed to becoming fully carbon neutral by 2030. In our model, the company cooperates with Amazon and uses Amazon's DDM advantages to attract many new consumers to buy their products, which supports the realization of carbon emission reduction goals.

Both government subsidies and the platform economy with DDM activities actively affect the improvement of the operational efficiency of the green supply chain. Based on this model, we ask three questions: (1) with government subsidies for green production, promotion, and consumption respectively, how do consumers' green preference behaviors affect the operation efficiency of the platform supply chain, in terms of the manufacturer's R&D and the third-party platform's DDM activities? (2) How does the level of government subsidies given to the manufacturer, the third-party platform, and consumer activities affect the green operation efficiency of the platform supply chain, and which subsidy strategy is more conducive to achieving a win-win situation of economic and ecological benefits? (3) With a combination of different selling models and subsidy policies, which strategy is more conducive to improving the supply chain's operational efficiency?

In order to answer the above questions, we constructed a platform supply chain consisting of a manufacturer and a third-party platform. The manufacturer produces green products via green R&D activities, while the third-party platform invests in DDM activities to promote the selling of these green products. For this paper, we developed the platform supply chain models on the assumption that the government subsidizes the manufacturer's green R&D, the third-party platform's DDM, and consumers' green consumption, respectively; we also compared the influence of three government subsidy policies on each member's decision and profits. We also study the preferred selling strategies of the manufacturer and the third-party platform in the platform supply chain by considering three government subsidy policies. In terms of a situation where the manufacturer sells green products to the third-party platform in an agency selling strategy, a similar reselling strategy is studied. Then, numerical examples are given to analyze the government subsidy policy and the online selling strategy. Our theoretical analysis shows that: (1) with the increase of the sensitivity coefficient of consumers to green R&D and DDM activities, the

profits of all members and the green R&D level of products also increase. That is, the more aware consumers are of green R&D and DDM activities, the more beneficial it is for all members in terms of obtaining more economical and ecological benefits meanwhile. (2) Under the agency selling and reselling strategies, the government subsidy strategy means that the manufacturer and the third-party platform can obtain maximum economic benefits, as well as improve the ecological benefits, depending on the government subsidy level, for each member in the platform supply chain. (3) The level of different government subsidies and the unit service commission fee from selling products lead the manufacturer and the third-party platform to choose their preferred selling strategies to achieve equilibrium in the selling strategy negotiated by the manufacturer and the third-party platform.

The rest of this paper is organized as follows. In Section 2, we review the relevant literature streams to identify the research gaps and our contributions. In Section 3, we describe the platform supply chain system and propose numerous related assumptions and symbols. In Section 4, the equilibrium results of six platform supply chains, with three kinds of government subsidy policies and two selling strategies, are analyzed using the backward induction method. We analyze the government subsidy policy and online selling strategy in Section 5. In Section 6, we discuss our conclusions and offer further possible research points.

## 2. Literature Review

This paper is closely related to the literature streams of the green supply chain, a supply chain with a government subsidy, and the platform supply chain.

### 2.1. Green Supply Chain Management

Green supply chains can help manufacturers to gain more economic benefits, as well as improved ecological benefits, which has been proved by many scholars from an empirical perspective. Chen et al. [17] explored the impact of green R&D on manufacturer operations and how to improve the level of enterprises' green R&D to cope with increasingly strict environmental protection requirements. Zhang et al. [18] showed that manufacturers can achieve more profits by producing green products alongside traditional products by the improvement of green consumption awareness. Several scholars have also explored the impact of consumer behavior on green supply chains through a numerical example analysis. Yu et al. [19] developed an optimization model under oligopolistic competition conditions, considering green preferences. The numerical results showed that the increase in consumer environmental awareness would incentivize manufacturers to produce more green products with higher green credentials. Zhu et al. [20] analyzed the selling of green products using different supply-chain structures in terms of manufacturers who produce green products and traditional products at the same time. The results show that the improvement of consumers' environmental awareness can promote the selling of green products within different supply chain structures. Zhu et al. [21] established a game model of cost-sharing contracts in the green supply chain, while numerical analysis showed that when consumers are more sensitive to the level of sustainability in green products, suppliers and producers can both enjoy greater profits. Hong et al. [22] discussed product design as part of the two-level green supply chain, using consumers' green consciousness as a reference point; the results showed that consumers' green consciousness plays a positive role in green R&D. Environmental sustainability has become an important metric for assessing the success of supply chain management. By studying the green supply chain's cooperation contract and its environmental performance, cooperation among supply chain members can help the green supply chain achieve better sustainability. Hong and Guo [23] studied the green supply chain cooperation contract and its environmental performance, then concluded that cooperation among supply chain members can help the supply chain to improve ecological efficiency; this will increase with improvement in the cooperation level of the members. In order to achieve sustainable growth and industrial upgrading, it is essential for the manufacturer to provide green improved products. Therefore, Yi et al. [24] studied

optimal strategies for the manufacturer and the retailer to sell green products under four contract conditions. The results showed that an increasing awareness of the environment and quality can help manufacturers and retailers to gain higher profits and the supply chain to gain higher ecological efficiency. Gao et al. [25] focused on green products using two different green technologies according to the government's green standards. The authors found that the improvement of green R&D technology can continuously improve the environmental benefits of the green supply chain.

The existing studies about the green supply chain mainly focus on a traditional supply chain and use empirical and modeling analysis methods. These results show that with an increase in consumers' green awareness, the manufacturer's green R&D activities can increase the sales of green products and improve the manufacturer's economic profits. In addition, the manufacturer's green R&D activities can reduce the greenhouse gases and hazardous wastes produced by the production of traditional products, so as to better achieve the goals of carbon emission reduction and carbon neutrality.

### 2.2. Supply Chain Management with Government Subsidy

Green R&D activities by manufacturers inevitably incur more costs, but the government can subsidize the supply chain members to motivate them to obtain more profits, which has been proved by certain scholars [26]. Yu et al. [19] established an oligopoly competition optimization model, considering green preferences and subsidy, with the goal of maximizing manufacturer profits. The results showed that an appropriate subsidy policy can not only generate more profits for the manufacturer but can also save subsidy investment for the government. Yang et al. [27] constructed a green supply chain model with a government subsidy under uncertain conditions, to study the influence of government subsidies on green manufacturing strategy. They believed that government subsidy measures can effectively stimulate the development of green products and increase the supply chain members' profits, as well as improve ecological efficiency. Xue et al. [28] found that the government subsidy could significantly improve ecological benefits and increase the selling of products by analyzing the decision-making regarding energy-saving products in the green supply chain. Different government subsidy strategies have different impacts on the operational efficiency of the supply chain. Yu et al. [29] discuss the effect of different government subsidy plans on the equilibrium strategy of enterprises, along with their ecological benefits. They found that government subsidies for consumers can better improve the ecological benefits. Bian et al. [30] compared the ecological benefits in terms of consumer or manufacturer subsidy, respectively, and concluded that government subsidies to consumers generated greater ecological benefits than subsidies to the manufacturer because the former can bring larger quantities of supplies and profits to the manufacturer. Sinayi and Rasti-Barzoki [31] concluded that government subsidy policies have a significant impact on the profits of supply chain members and the environment via studying the optimal decision-making of green supply chain members with or without government subsidy. The government subsidizes the manufacturer's technology to produce green products and can effectively help the manufacturer to produce green products, which are more popular with consumers and are more environmentally friendly [32–34]. Bi et al. [35] proposed that when consumers are environmentally conscious, the government can use subsidy policies to motivate manufacturers to adopt green emission reduction technologies. Jung et al. [36] researched government subsidies for the development of green technology, its impact on the environment, and ecological benefits. The study showed that when the government provides subsidies for the development of green technology, the ecological benefits can be effectively improved. Li et al. [37] studied the impact of government subsidies on the innovation level of the secondary supply chain. They found that government subsidies to consumers can better stimulate the manufacturer to carry out green innovation.

The above studies show that government subsidies are beneficial for supply chain members to obtain more economic benefits, and stimulate manufacturers to produce more

green products, thereby improving the ecological benefits. Moreover, the government can improve the operation efficiency of the supply chain by subsidizing different members of the supply chain system.

*2.3. Platform Supply Chain Management*

The addition of the platform can improve the operational efficiency of the supply chain and help supply-chain members to save costs and gain more economic benefits. Li [38] established a mode that takes cost as a variable of technology investment decisions in the platform supply chain. Li compared the traditional supply chain and the platform supply chain, then found that even if the third-party platform does not bear any part of the channel costs, it can invest in reducing the channel costs. Wang et al. [39] analyzed the panel data of 7309 e-retailers and proved that manufacturers selling products on e-commerce platforms have a first-mover advantage. Wang [40] investigates the decision-making, coordination contract, and altruistic preference of an e-commerce supply chain composed of a manufacturer and a third-party platform. The research showed that the platform's altruistic preferences could help the manufacturer to increase profits. Other scholars introduced greening into the platform supply chain and found that the platform can help manufacturers to improve the economic benefits as well as improve environmental sustainability. Qi [41] proposed that green modular design has the risk of losing the product platform planning strategy. To solve this problem, Qi built a design method that considered both the production environment and platform planning strategy, then verified the effectiveness of the method using numerical example analysis. The platform and the manufacturer usually adopted the agency selling and reselling model proposed by Hagiu and Wright [12]. Abhishek et al. [42] answered a key question that e-tailers are facing: whether to choose an agency selling model or a traditional reselling model. They used theoretical model analysis to show that the retailers' choice of selling model depends on the competition among them. However, under specific conditions, choosing the agency-selling model is beneficial to the supply chain members. Geng [43] discussed the upstream manufacturer and downstream online platform with additional product pricing, in terms of the interaction between selling mode selections, and found that the selling model affected the choice between additional pricing and bundled pricing. When the commission rate is not too small and the market potential of additional products on the platform is not large, the platform is more inclined to accept agency selling. The platform can provide the manufacturer with a new market competitive advantage. Through data analysis technology, the platform can effectively help the manufacturer to increase product market selling, gain more profits, and take advantage of being the first mover. For example, the platform supply chain's DDM has become a hot topic in recent years. The platform supply chain can describe, predict, analyze, and guide consumer behavior, based on data-driven analysis; that is, by data-driven marketing. Therefore, Braverman [13] proposed that data-driven marketing technology can provide the manufacturer with better marketing activities for green products and further enable the manufacturer to obtain more economic benefits. Liu et al. [44] considered the influence of DDM and investigated the preferences of a platform between agency selling and reselling. To do this, they established and compared four models: NO-DDM + agency selling, NO-DDM + reselling, DDM+ agency selling, and DDM+ reselling. The results showed that DDM activities can help the manufacturer to increase the selling volume, bringing more profits.

It can be seen from the above studies that the platform's agency selling and reselling functions have different impacts on the operational efficiency of the platform supply chain. The third-party platform's DDM activities can help the manufacturer expand market demand to gain greater economic benefits.

*2.4. Research Gaps*

From the above literature review, we can see that there are abundant possibilities for growth in the green supply chain, a supply chain with a government subsidy, and the

platform supply chain. For example, Zhang et al. [18] considered that if the manufacturer produced green products while producing traditional products, they could then sell them through the traditional retail channels. Yang et al. [27] constructed a supply chain model including the government subsidy activities; their research focus was on selling green products in the traditional retail supply chains. The platform economy has become a hot topic in the current research by scholars. However, the above studies have not considered marketing activities regarding green products in the platform supply chain. Liu et al. [44] considered that when the platform provided agency selling or reselling services, the platform uses its own DDM advantage to sell products. However, the manufacturer sells the traditional products rather than the green products through the platform, and they do not consider the government's subsidies for the green products. To sum up, there is little research on the selling of green products in the supply chain through the platform. There is as yet no research on government subsidies for green R&D activities on the platform supply chain to which the platform can provide agency selling or reselling services. Therefore, in order to bridge this research gap, we established a platform supply chain composed of a manufacturer and a third-party platform. Then, we analyzed the operational efficiency and the choice of selling strategies for each member of the platform supply chain when the government implemented subsidy strategies and the platform provided different selling services.

## 3. Model Development and Symbol Description

### 3.1. Model Description

This paper presents a platform supply chain system consisting of a manufacturer and a third-party platform. In this system, the manufacturer develops and produces green products, then sells them to consumers with green preferences through the third-party platform. The third-party platform can provide two kinds of selling strategies for the manufacturer; one is the agency selling strategy, that is, the manufacturer directly sells the green products to consumers through the third-party platform after paying the unit service commission fee for selling the products. Another is the reselling strategy, that is, the manufacturer wholesales green products to the third-party platform, then the third-party platform sells them to consumers at a marked-up price. In the process of selling products, the platform provides DDM activities to promote the selling of green products. The government provides three types of subsidy policies, one is to subsidize the manufacturer's green R&D, the second is to subsidize the third-party platform's DDM, and the third is to subsidize consumers' green consumption in order to encourage the platform supply chain members to implement green activities. The operation process of the platform supply chain under government subsidies is shown in Figure 1.

**Assumption 1.** *There is a Stackelberg game relationship between the manufacturer and the third-party platform; the manufacturer is the leader, and the third-party platform is the follower. They are entirely rational and show information symmetry with each other.*

**Assumption 2.** *The manufacturer determines the green R&D level, e. Because investments are often diseconomies of scale, we set the manufacturer's green R&D level cost at $e^2/2$ [41]. In order to analyze the impact of green R&D level, we normalize the unit production cost of green products as zero [45].*

**Assumption 3.** *(1) Under the agency selling strategy, the manufacturer directly sells green products to the third-party platform at the unit selling price, p, and needs to pay the third-party platform the unit service commission fee, u . (2) Under the reselling strategy, the manufacturer sells the green products to the third-party platform at a unit wholesale price w, then the third-party platform adds a marked-up price for green products, m, and sells them to consumers at the unit selling price, p; it is obvious that $p = m + w$. At the same time, the third-party platform uses the acquired consumer behavior information to conduct agency selling and reselling DDM activities. We set the DDM level at t because the technology investment cost is usually not economical in scale; thus, we set the third-party platform's DDM level cost at $t^2/2$ [46].*

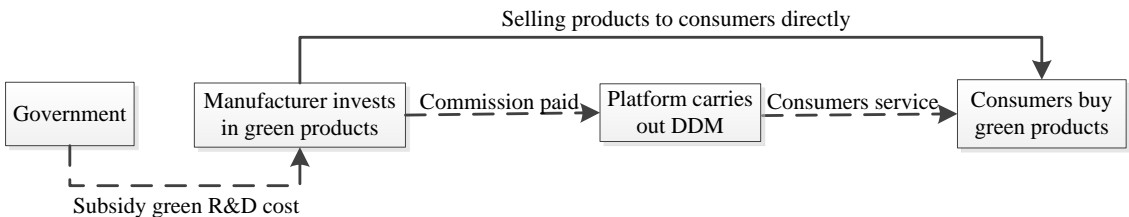

**Model 1:** Agency selling strategy of subsidizing the manufacturer's green R&D.

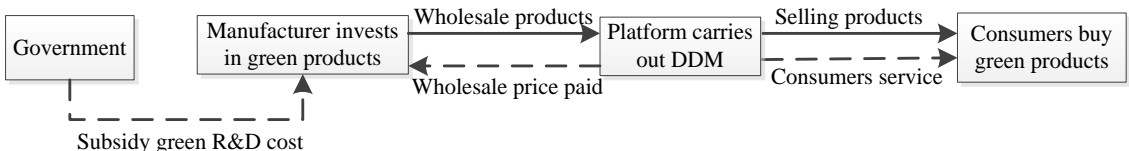

**Model 2:** Reselling strategy of subsidizing the manufacturer's green R&D.

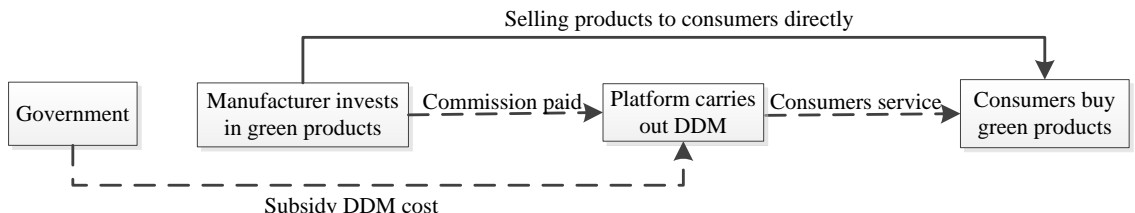

**Model 3:** Agency selling strategy of subsidizing the platform's DDM.

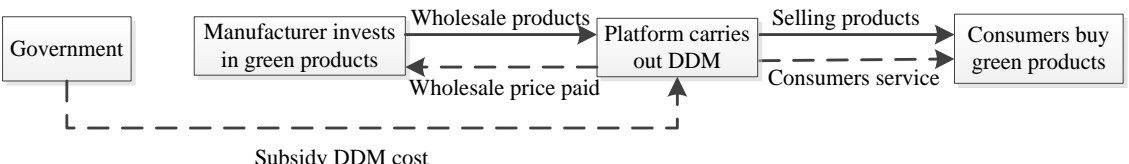

**Model 4:** Reselling strategy of subsidizing the platform's DDM.

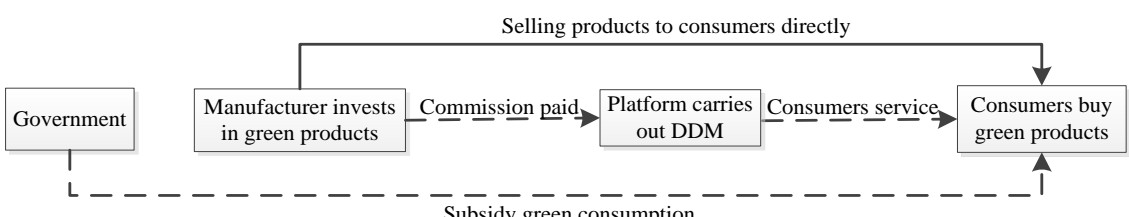

**Model 5:** Agency selling strategy of subsidizing consumers' green consumption.

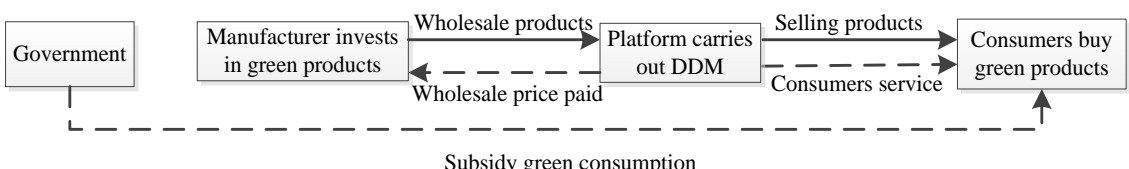

**Model 6:** Reselling strategy of subsidizing consumers' green consumption.

**Figure 1.** The operation process of the government subsidy for the platform supply chain under green R&D.

**Assumption 4.** *Consumers in the market have green preferences and the manufacturer's green R&D activities can increase their perception of their effects. If we refer to the results in certain references [47,48], the market demand function of green products meets the uniform distribution of* $[0,1]$ *and so the consumer surplus is* $V = \mu - p + \alpha e + \beta t$ *Consumers will buy products only when the consumer surplus is greater than zero, so we can achieve* $\mu > p - \alpha e - \beta t$ *and* $D = \int_{p-\alpha e-\beta t}^{1} 1 \mathrm{du} = 1 - p + \alpha e + \beta t$. *Therefore, the market demand function of green products is* $D = 1 - p + \alpha e + \beta t$. *Thus, the total products' service commission fee is* $uD$ *under the agency selling strategy. Among the variables, 1 is the potential market size,$\alpha$ is the sensitivity coefficient of consumers to green R&D activities, and $\beta$ is the sensitivity coefficient of consumers to DDM activities.*

**Assumption 5.** *Similar to reference [45], the government subsidies of the green activities of supply chain members depend on the related green costs or consumption. The government subsidizes the manufacturer's green R&D cost in proportion to* $\theta_e$, *the third-party platform's DDM cost in proportion to* $\theta_t$. *The government subsidies to green consumption are reflected in the consumer surplus of green products. When consumers buy green products, they can receive subsidy* $S_c$ *for each unit of green products from the government.*

*3.2. Symbol Description*

The variable symbols used in our work are as shown in Table 1.

**Table 1.** Notations.

| Endogenous Variables | Meanings |
|---|---|
| $D$ | Market demand for green products |
| $w$ | Unit wholesale price of green products |
| $m$ | Marked-up price of green products |
| $p$ | Unit selling price of green products |
| $e$ | Green R&D level |
| $t$ | DDM level |

| Exogenous Variables | Meanings |
|---|---|
| $\alpha$ | The sensitivity coefficient of consumers to green R&D activities |
| $\beta$ | The sensitivity coefficient of consumers to DDM activities |
| $u$ | Unit service commission fee for selling products |
| $\theta_e$ | The subsidy, proportional to green R&D cost |
| $\theta_t$ | The subsidy, proportional to the DDM cost |
| $S_c$ | The unit subsidy fee to consumers |

In this section, subscript $i \in \{M, P\}$ represents the manufacturer and the third-party platform, while superscript $j \in \{a, r\}$ represents the agency selling strategy and the reselling strategy. The superscript $*$ represents the equilibrium solution and profits.

**4. Model Development**

*4.1. Agency-Selling Models*

Under the agency-selling strategy, the operation process of the platform supply chain is as follows: the manufacturer directly sells green products to consumers through the third-party platform, after paying the unit service commission fee for selling products.

4.1.1. R&D Subsidy Model

The government subsidizes the manufacturer's green R&D cost with the proportion of $\theta_e$; the profits functions of the manufacturer and the third-party platform are as follows:

$$\Pi_M^{a-\theta_e}(p, e) = (p - u)D - (1 - \theta_e)\frac{e^2}{2} \tag{1}$$

$$\Pi_P^{a-\theta_e}(t) = uD - \frac{t^2}{2} \tag{2}$$

The equilibrium decisions and profits of the manufacturer and the third-party platform can be obtained as shown in proposition 1 with the backward induction method.

**Proposition 1.** *When the government subsidizes the manufacturer's green R&D, the equilibrium results of the platform supply chain with an agency selling strategy are as follows:*
$D^{a-\theta_e*} = \frac{(1-\theta_e)[1-u(1-\beta^2)]}{2(1-\theta_e)-\alpha^2}$, $p^{a-\theta_e*} = \frac{[(1+\beta^2)(1-\theta_e)-\alpha^2]u+(1-\theta_e)}{2(1-\theta_e)-\alpha^2}$, $e^{a-\theta_e*} = \frac{\alpha[1-u(1-\beta^2)]}{2(1-\theta_e)-\alpha^2}$,
$t^{a-\theta_e*} = \beta u$, $\Pi_M^{a-\theta_e*} = \frac{(1-\theta_e)[1-u(1-\beta^2)]^2}{2[2(1-\theta_e)-\alpha^2]}$, $\Pi_P^{a-\theta_e*} = \frac{u[u\alpha^2\beta^2+2(1-\theta_e)(1-u)]}{2[2(1-\theta_e)-\alpha^2]}$.

### 4.1.2. DDM Subsidy Model

The government subsidizes the third-party platform's DDM cost with a proportion of $\theta_t$; the profits functions of the manufacturer and the third-party platform are as follows:

$$\Pi_M^{a-\theta_t}(p,e) = (p-u)D - \frac{e^2}{2} \tag{3}$$

$$\Pi_P^{a-\theta_t}(t) = uD - (1-\theta_t)\frac{t^2}{2} \tag{4}$$

The equilibrium decisions and profits of the manufacturer and the third-party platform can be obtained as shown in Proposition 2 with the backward induction method.

**Proposition 2.** *When the government subsidizes the third-party platform's DDM, the equilibrium results of the platform supply chain with agency selling strategy are as follows:*
$D^{a-\theta_t*} = \frac{\beta^2 u+(1-\theta_t)(1-u)}{(2-\alpha^2)(1-\theta_t)}$, $p^{a-\theta_t*} = \frac{[1+u(1-\alpha^2)](1-\theta_t)+\beta^2 u}{(2-\alpha^2)(1-\theta_t)}$, $e^{a-\theta_t*} = \frac{\alpha[\beta^2 u+(1-\theta_t)(1-u)]}{(2-\alpha^2)(1-\theta_t)}$,
$t^{a-\theta_t} = \frac{\beta u}{1-\theta_t}$, $\Pi_M^{a-\theta_t*} = \frac{[1-(1-u)\theta_t-u(1-\beta^2)]^2}{2(2-\alpha^2)(1-\theta_t)^2}$, $\Pi_P^{a-\theta_t*} = \frac{u[u\alpha^2\beta^2+2(1-\theta_t)(1-u)]}{2(2-\alpha^2)(1-\theta_t)}$.

### 4.1.3. Consumption Subsidy Model

The market demand function of green products meets the uniform distribution of $[0,1]$; when the government subsidizes consumers' green consumption with the unit subsidy fee $S_c$, the consumer surplus is $V = \mu - p + \alpha e + \beta t + S_c$. Consumers will buy products only when the consumer surplus is greater than zero, so we can achieve $\mu > p - \alpha e - \beta t - S_c$ and $D = \int_{p-\alpha e-\beta t-S_c}^1 1du = 1 - p + \alpha e + \beta t + S_c$. The market demand function of green products is $D = 1 - p + \alpha e + \beta t + S_c$. At this time, the profits functions of the manufacturer and the third-party platform are as follows:

$$\Pi_M^{a-s_c}(p,e) = (p-u)D - \frac{e^2}{2} \tag{5}$$

$$\Pi_P^{a-s_c}(t) = uD - \frac{t^2}{2}. \tag{6}$$

The equilibrium decisions and profits of the manufacturer and the third-party platform can be obtained as shown in Proposition 3 with the backward induction method.

**Proposition 3.** *When the government subsidizes consumers' green consumption, the equilibrium results of the platform supply chain with agency selling strategy are as follows:*
$D^{a-s_c*} = \frac{1+s_C-u(1-\beta^2)}{2-\alpha^2}$, $p^{a-s_c*} = \frac{[1+s_C+u(1-\alpha^2+\beta^2)]}{2-\alpha^2}$, $e^{a-s_c*} = \frac{\alpha[1+s_C-u(1-\beta^2)]}{2-\alpha^2}$, $t^{a-s_c*} = \beta u$,
$\Pi_M^{a-s_c*} = \frac{[1+s_C-u(1-\beta^2)]^2}{2(2-\alpha^2)}$, $\Pi_P^{a-s_c*} = \frac{u[\alpha^2\beta^2 u+2(1-u+s_c)]}{2(2-\alpha^2)}$.

### 4.2. Reselling Models

Under the reselling strategy, the operation process of the platform supply chain is as follows: the manufacturer resells the green products to the third-party platform at the unit wholesale price. The third-party platform sells them to consumers at a marked-up price.

#### 4.2.1. R&D Subsidy Model

The government subsidizes the manufacturer's green R&D cost with a proportion of $\theta_e$, the profits functions of the manufacturer and the third-party platform are as follows:

$$\Pi_M^{r-\theta_e}(w,e) = wD - (1-\theta_e)\frac{e^2}{2} \tag{7}$$

$$\Pi_P^{r-\theta_e}(m,t) = mD - \frac{t^2}{2} \tag{8}$$

The equilibrium decisions and profits of the manufacturer and the third-party platform can be obtained as shown in Proposition 4 with the backward induction method.

**Proposition 4** . *When the government subsidizes the manufacturer's green R&D, the equilibrium results of the platform supply chain with reselling strategy are as follows:*
$D^{r-\theta_e*} = \frac{1-\theta_e}{2(2-\beta^2)(1-\theta_e)-\alpha^2}$, $w^{r-\theta_e*} = \frac{(2-\beta^2)(1-\theta_e)}{2(2-\beta^2)(1-\theta_e)-\alpha^2}$, $m^{r-\theta_e*} = \frac{1-\theta_e}{2(2-\beta^2)(1-\theta_e)-\alpha^2}$, $p^{r-\theta_e*} = \frac{(3-\beta^2)(1-\theta_e)}{2(2-\beta^2)(1-\theta_e)-\alpha^2}$, $e^{r-\theta_e*} = \frac{\alpha}{2(2-\beta^2)(1-\theta_e)-\alpha^2}$, $t^{r-\theta_e*} = \frac{\beta(1-\theta_e)}{2(2-\beta^2)(1-\theta_e)-\alpha^2}$, $\Pi_M^{r-\theta_e*} = \frac{1-\theta_e}{2[2(2-\beta^2)(1-\theta_e)-\alpha^2]}$, $\Pi_P^{r-\theta_e*} = \frac{(2-\beta^2)(1-\theta_e)^2}{2[2(2-\beta^2)(1-\theta_e)-\alpha^2]^2}$.

#### 4.2.2. DDM Subsidy Model

The government subsidizes the third-party platform's DDM cost at the proportion of $\theta_t$; the profits functions of the manufacturer and the third-party platform are as follows:

$$\Pi_M^{r-\theta_t}(w,e) = wD - \frac{e^2}{2} \tag{9}$$

$$\Pi_P^{r-\theta_t}(m,t) = mD - (1-\theta_t)\frac{t^2}{2}. \tag{10}$$

The equilibrium decisions and profits of the manufacturer and the third-party platform can be obtained as shown in Proposition 5 with the backward induction method.

**Proposition 5.** *When the government subsidizes the third-party platform's DDM, the equilibrium results of the platform supply chain with reselling strategy are as follows:*
$D^{r-\theta_t*} = \frac{1-\theta_t}{(4-\alpha^2)(1-\theta_t)-2\beta^2}$, $w^{r-\theta_t*} = \frac{2(1-\theta_t)-\beta^2}{(4-\alpha^2)(1-\theta_t)-2\beta^2}$, $m^{r-\theta_t*} = \frac{1-\theta_t}{(4-\alpha^2)(1-\theta_t)-2\beta^2}$, $p^{r-\theta_t*} = \frac{3(1-\theta_t)-\beta^2}{(4-\alpha^2)(1-\theta_t)-2\beta^2}$, $e^{r-\theta_t*} = \frac{\alpha(1-\theta_t)}{(4-\alpha^2)(1-\theta_t)-2\beta^2}$, $t^{r-\theta_t*} = \frac{\beta}{(4-\alpha^2)(1-\theta_t)-2\beta^2}$, $\Pi_M^{r-\theta_t*} = \frac{1-\theta_t}{2[(4-\alpha^2)(1-\theta_t)-2\beta^2]}$, $\Pi_P^{r-\theta_t} = \frac{(1-\theta_t)[2(1-\theta_t)-\beta^2]}{2[(4-\alpha^2)(1-\theta_t)-2\beta^2]^2}$.

#### 4.2.3. Consumption Subsidy Model

Similar to Section 4.1.3, the unit subsidy fee to consumers is $S_c$; the profits functions of the manufacturer and the third-party platform are as follows:

$$\Pi_M^{r-s_c}(w,e) = wD - \frac{e^2}{2} \tag{11}$$

$$\Pi_P^{r-s_c}(m,t) = mD - \frac{t^2}{2} \tag{12}$$

The equilibrium decisions and profits of the manufacturer and the third-party platform can be obtained as shown in Proposition 6 with the backward induction method.



**Proposition 6.** *When the government subsidizes consumers' green consumption, the equilibrium results of the platform supply chain with reselling strategy are as follows:* $D^{r-s_c*} = \frac{1+s_C}{4-\alpha^2-2\beta^2}$,

$w^{r-s_c*} = \frac{(1+s_C)(2-\beta^2)}{4-\alpha^2-2\beta^2}$, $m^{r-s_c*} = \frac{1+s_C}{4-\alpha^2-2\beta^2}$, $p^{r-s_c*} = \frac{(1+s_C)(3-\beta^2)}{4-\alpha^2-2\beta^2}$, $e^{r-s_c*} = \frac{\alpha(s_C+1)}{4-\alpha^2-2\beta^2}$,

$t^{r-s_c*} = \frac{\beta(1+s_C)}{4-\alpha^2-2\beta^2}$, $\Pi_M^{r-s_c*} = \frac{(1+s_C)^2}{2(4-\alpha^2-2\beta^2)}$, $\Pi_P^{r-s_c*} = \frac{(2-\beta^2)(1+s_C)^2}{2(4-\alpha^2-2\beta^2)^2}$.

Please see Appendix A for the detailed proof process of Propositions 1–6.

*4.3. Sensitivity Analysis*

**Conclusion 1.** $\frac{\partial D^{a-\theta_e*}}{\partial \alpha} > 0$, $\frac{\partial p^{a-\theta_e*}}{\partial \alpha} > 0$, $\frac{\partial e^{a-\theta_e*}}{\partial \alpha} > 0$, $\frac{\partial t^{a-\theta_e*}}{\partial \alpha} = 0$, $\frac{\partial \Pi_M^{a-\theta_e*}}{\partial \alpha} > 0$, $\frac{\partial \Pi_P^{a-\theta_e*}}{\partial \alpha} > 0$;

$\frac{\partial D^{a-\theta_t*}}{\partial \alpha} > 0$, $\frac{\partial p^{a-\theta_t*}}{\partial \alpha} > 0$, $\frac{\partial e^{a-\theta_t*}}{\partial \alpha} > 0$, $\frac{\partial t^{a-\theta_t*}}{\partial \alpha} = 0$, $\frac{\partial \Pi_M^{a-\theta_t*}}{\partial \alpha} > 0$, $\frac{\partial \Pi_P^{a-\theta_t*}}{\partial \alpha} > 0$; $\frac{\partial D^{a-s_c*}}{\partial \alpha} > 0$,

$\frac{\partial p^{a-s_c*}}{\partial \alpha} > 0$, $\frac{\partial e^{a-s_c*}}{\partial \alpha} > 0$, $\frac{\partial t^{a-s_c*}}{\partial \alpha} = 0$, $\frac{\partial \Pi_M^{a-s_c*}}{\partial \alpha} > 0$, $\frac{\partial \Pi_P^{a-s_c*}}{\partial \alpha} > 0$; $\frac{\partial D^{r-\theta_e*}}{\partial \alpha} > 0$, $\frac{\partial w^{r-\theta_e*}}{\partial \alpha} > 0$,

$\frac{\partial m^{r-\theta_e*}}{\partial \alpha} > 0$, $\frac{\partial p^{r-\theta_e*}}{\partial \alpha} > 0$, $\frac{\partial e^{r-\theta_e*}}{\partial \alpha} > 0$, $\frac{\partial t^{r-\theta_e*}}{\partial \alpha} > 0$, $\frac{\partial \Pi_M^{r-\theta_e*}}{\partial \alpha} > 0$, $\frac{\partial \Pi_P^{r-\theta_e*}}{\partial \alpha} > 0$. $\frac{\partial D^{r-\theta_t*}}{\partial \alpha} > 0$,

$\frac{\partial w^{r-\theta_t*}}{\partial \alpha} > 0$, $\frac{\partial m^{r-\theta_t*}}{\partial \alpha} > 0$, $\frac{\partial p^{r-\theta_t*}}{\partial \alpha} > 0$, $\frac{\partial e^{r-\theta_t*}}{\partial \alpha} > 0$, $\frac{\partial t^{r-\theta_t*}}{\partial \alpha} > 0$, $\frac{\partial \Pi_M^{r-\theta_t*}}{\partial \alpha} > 0$, $\frac{\partial \Pi_P^{r-\theta_t*}}{\partial \alpha} > 0$;

$\frac{\partial D^{r-s_c*}}{\partial \alpha} > 0$, $\frac{\partial w^{r-s_c*}}{\partial \alpha} > 0$, $\frac{\partial m^{r-s_c*}}{\partial \alpha} > 0$, $\frac{\partial p^{r-s_c*}}{\partial \alpha} > 0$, $\frac{\partial e^{r-s_c*}}{\partial \alpha} > 0$, $\frac{\partial t^{r-s_c*}}{\partial \alpha} > 0$, $\frac{\partial \Pi_M^{r-s_c*}}{\partial \alpha} > 0$,

$\frac{\partial \Pi_P^{r-s_c*}}{\partial \alpha} > 0$.

Conclusion 1 shows that when the sensitivity coefficient of consumers to green R&D activities increases, more and more consumers are going to buy green products; therefore, the manufacturer increases the unit selling price of the green products. In addition, it also encourages the manufacturer to improve the green R&D level; then, more potential consumers are willing to buy green products. At this point, the increase in market demand and the unit selling price of green products will bring more revenue to the manufacturer than the increase in green R&D cost, so the manufacturer's profit increases. Under the agency-selling strategy, the sensitivity coefficient of consumers to green R&D activities does not affect the DDM level. At this point, the market demand for green products increases the unit service commission fee of products; therefore, the third-party platform's profit increases. Under the reselling strategy, as the sensitivity coefficient of consumers to green R&D activities increases, the third-party platform is encouraged to improve the DDM level, which also increases the market demand for green products. The increase in the marked-up price and the unit selling price of green products will bring more revenue to the third-party platform than the increase in DDM cost. Therefore, the third-party platform's profit increases.

**Conclusion 2.** $\frac{\partial D^{a-\theta_e*}}{\partial \beta} > 0$, $\frac{\partial p^{a-\theta_e*}}{\partial \beta} > 0$, $\frac{\partial e^{a-\theta_e*}}{\partial \beta} > 0$, $\frac{\partial t^{a-\theta_e*}}{\partial \beta} > 0$, $\frac{\partial \Pi_M^{a-\theta_e*}}{\partial \beta} > 0$, $\frac{\partial \Pi_P^{a-\theta_e*}}{\partial \beta} > 0$;

$\frac{\partial D^{a-\theta_t*}}{\partial \beta} > 0$, $\frac{\partial p^{a-\theta_t*}}{\partial \beta} > 0$, $\frac{\partial e^{a-\theta_t*}}{\partial \beta} > 0$, $\frac{\partial t^{a-\theta_t*}}{\partial \beta} > 0$, $\frac{\partial \Pi_M^{a-\theta_t*}}{\partial \beta} > 0$, $\frac{\partial \Pi_P^{a-\theta_t*}}{\partial \beta} > 0$; $\frac{\partial D^{a-s_c*}}{\partial \beta} > 0$,

$\frac{\partial p^{a-s_c*}}{\partial \beta} > 0$, $\frac{\partial e^{a-s_c*}}{\partial \beta} > 0$, $\frac{\partial t^{a-s_c*}}{\partial \beta} > 0$, $\frac{\partial \Pi_M^{a-s_c*}}{\partial \beta} > 0$, $\frac{\partial \Pi_P^{a-s_c*}}{\partial \beta} > 0$. $\frac{\partial D^{r-\theta_e*}}{\partial \beta} > 0$, $\frac{\partial w^{r-\theta_e*}}{\partial \beta} > 0$,

$\frac{\partial m^{r-\theta_e*}}{\partial \beta} > 0$, $\frac{\partial p^{r-\theta_e*}}{\partial \beta} > 0$, $\frac{\partial e^{r-\theta_e*}}{\partial \beta} > 0$, $\frac{\partial t^{r-\theta_e*}}{\partial \beta} > 0$, $\frac{\partial \Pi_M^{r-\theta_e*}}{\partial \beta} > 0$, $\frac{\partial \Pi_P^{r-\theta_e*}}{\partial \beta} > 0$; $\frac{\partial D^{r-\theta_t*}}{\partial \beta} > 0$,

$\frac{\partial w^{r-\theta_t*}}{\partial \beta} > 0$, $\frac{\partial m^{r-\theta_t*}}{\partial \beta} > 0$, $\frac{\partial p^{r-\theta_t*}}{\partial \beta} > 0$, $\frac{\partial e^{r-\theta_t*}}{\partial \beta} > 0$, $\frac{\partial t^{r-\theta_t*}}{\partial \beta} > 0$, $\frac{\partial \Pi_M^{r-\theta_t*}}{\partial \beta} > 0$, $\frac{\partial \Pi_P^{r-\theta_t*}}{\partial \beta} > 0$;

$\frac{\partial D^{r-s_c*}}{\partial \beta} > 0$, $\frac{\partial w^{r-s_c*}}{\partial \beta} > 0$, $\frac{\partial m^{r-s_c*}}{\partial \beta} > 0$, $\frac{\partial p^{r-s_c*}}{\partial \beta} > 0$, $\frac{\partial e^{r-s_c*}}{\partial \beta} > 0$, $\frac{\partial t^{r-s_c*}}{\partial \beta} > 0$, $\frac{\partial \Pi_M^{r-s_c*}}{\partial \beta} > 0$,

$\frac{\partial \Pi_P^{r-s_c*}}{\partial \beta} > 0$.

Conclusion 2 shows that when the sensitivity coefficient of consumers to the DDM activities increases, more and more consumers are going to buy green products; therefore, the manufacturer increases the unit selling price of the green products. In addition, it also encourages the manufacturer to improve green R&D levels; therefore, more potential consumers are willing to buy green products. At this point, the increase in the market demand and the unit selling price of green products bring more revenue to the manufacturer

than the increase in green R&D costs; therefore, the manufacturer's profit increases. At the same time, as the sensitivity coefficient of consumers to DDM activities increases, the third-party platform is encouraged to improve the DDM level, which also promotes the increase in the market demand for green products. With the increase in the unit selling price of green products, the increase in the unit service commission fee of selling products will bring more revenue to the third-party platform than the increase in DDM cost. Therefore, the third-party platform's profit increases.

**Conclusion 3.** $\frac{\partial D^{a-\theta_e*}}{\partial \theta_e} > 0$, $\frac{\partial p^{a-\theta_e*}}{\partial \theta_e} > 0$, $\frac{\partial e^{a-\theta_e*}}{\partial \theta_e} > 0$, $\frac{\partial t^{a-\theta_e*}}{\partial \theta_e} = 0$, $\frac{\partial \Pi_M^{a-\theta_e*}}{\partial \theta_e} > 0$, $\frac{\partial \Pi_P^{a-\theta_e*}}{\partial \theta_e} > 0$. $\frac{\partial D^{r-\theta_e*}}{\partial \theta_e} > 0$, $\frac{\partial w^{r-\theta_e*}}{\partial \theta_e} > 0$, $\frac{\partial m^{r-\theta_e*}}{\partial \theta_e} > 0$, $\frac{\partial p^{r-\theta_e*}}{\partial \theta_e} > 0$, $\frac{\partial e^{r-\theta_e*}}{\partial \theta_e} > 0$, $\frac{\partial t^{r-\theta_e*}}{\partial \theta_e} > 0$, $\frac{\partial \Pi_M^{r-\theta_e*}}{\partial \theta_e} > 0$, $\frac{\partial \Pi_P^{r-\theta_e*}}{\partial \theta_e} > 0$.

Conclusion 3 shows that the increase in the subsidy proportion to green R&D cost encourages the manufacturer to improve its green R&D level; therefore, more potential consumers are willing to buy green products, which increases the unit selling price of green products. The increase in the market demand and the unit selling price of these green products will bring more revenue to the manufacturer than the increase in green R&D cost; therefore, the manufacturer's profit increases. Under the agency-selling strategy, the subsidy proportional to green R&D costs does not affect the DDM level. At this point, the market demand for green products increases the unit service commission fee of selling products, so the third-party platform's profit increases. Under the reselling strategy, the increase in the market demand and the unit selling price of green products brings more revenue to the third-party platform than the increase in the DDM cost; therefore, the third-party platform's profit increases.

**Conclusion 4.** $\frac{\partial D^{a-\theta_t*}}{\partial \theta_t} > 0$, $\frac{\partial p^{a-\theta_t*}}{\partial \theta_t} > 0$, $\frac{\partial e^{a-\theta_t*}}{\partial \theta_t} > 0$, $\frac{\partial t^{a-\theta_t*}}{\partial \theta_t} > 0$, $\frac{\partial \Pi_M^{a-\theta_t*}}{\partial \theta_t} > 0$, $\frac{\partial \Pi_P^{a-\theta_t*}}{\partial \theta_t} > 0$. $\frac{\partial D^{r-\theta_t*}}{\partial \theta_t} > 0$, $\frac{\partial w^{r-\theta_t*}}{\partial \theta_t} > 0$, $\frac{\partial m^{r-\theta_t*}}{\partial \theta_t} > 0$, $\frac{\partial p^{r-\theta_t*}}{\partial \theta_t} > 0$, $\frac{\partial e^{r-\theta_t*}}{\partial \theta_t} > 0$, $\frac{\partial t^{r-\theta_t*}}{\partial \theta_t} > 0$, $\frac{\partial \Pi_M^{r-\theta_t*}}{\partial \theta_t} > 0$, $\frac{\partial \Pi_P^{r-\theta_t*}}{\partial \theta_t} > 0$.

Conclusion 4 shows that the increase in the subsidy proportional to DDM cost encourages the third-party platform to improve its DDM level, after which more potential consumers are willing to buy green products. The increase in the market demand and the unit selling price of green products brings more revenue to the third-party platform than the increase in the DDM cost; therefore, the third-party platform's profit increases. The higher subsidy proportional to DDM cost encourages the manufacturer to improve the green R&D level, in order to attract more potential consumers to buy green products. The increased unit service commission fee for selling products causes the manufacturer to raise the unit selling price of green products. Finally, the manufacturer's profit increases.

**Conclusion 5.** $\frac{\partial D^{a-s_c*}}{\partial s_c} > 0$, $\frac{\partial p^{a-s_c*}}{\partial s_c} > 0$, $\frac{\partial e^{a-s_c*}}{\partial s_c} > 0$, $\frac{\partial t^{a-s_c*}}{\partial s_c} = 0$, $\frac{\partial \Pi_M^{a-s_c*}}{\partial s_c} > 0$, $\frac{\partial \Pi_P^{a-s_c*}}{\partial s_c} > 0$. $\frac{\partial D^{r-s_c*}}{\partial s_c} > 0$, $\frac{\partial w^{r-s_c*}}{\partial s_c} > 0$, $\frac{\partial m^{r-s_c*}}{\partial s_c} > 0$, $\frac{\partial p^{r-s_c*}}{\partial s_c} > 0$, $\frac{\partial e^{r-s_c*}}{\partial s_c} > 0$, $\frac{\partial t^{r-s_c*}}{\partial s_c} > 0$, $\frac{\partial \Pi_M^{r-s_c*}}{\partial s_c} > 0$, $\frac{\partial \Pi_P^{r-s_c*}}{\partial s_C} > 0$.

Conclusion 5 shows that when the unit subsidy fee to consumers increases, more potential consumers are willing to buy green products, which encourages the manufacturer to improve the green R&D level. The unit selling price of green products increases the manufacturer's profit. Under the agency selling strategy, the unit subsidy fee to consumers does not affect the DDM level. Under the reselling strategy, the increase in the unit subsidy fee to consumers encourages the third-party platform to improve the DDM level. As the market demand and the unit selling price of green products increase, the third-party platform will charge more unit subsidy fees for selling products, so the third-party platform's profit increases.

## 5. Numerical Analysis

We examined the optimal government subsidy policy and online selling strategy in the platform supply chain with green R&D and DDM activities, using numerical analysis. In this section, we use $\alpha = 0.3$, $\beta = 0.4$. $\{a, r\}$ to represent the agency selling and reselling strategy. $\theta_e$, $\theta_t$ and $S_c$, respectively, represent the government subsidies for the green activities of the manufacturer, the third-party platform and consumers. For example, $\{a - \theta_e\}$ represents the agency-selling strategy of the government subsidizing the manufacturer's green R&D. The same applies to others.

### 5.1. The Government Subsidy Policy

How to better realize the policy effect under the same subsidy level is a problem that the government should consider. Therefore, we assume that the subsidy is proportional to green R&D cost, DDM cost, and the unit subsidy fee, assuming the consumers are the same, in order to explore which subsidy policy can obtain good ecological and economic benefits. From Figure 2, we can achieve that under the agency selling strategy:

**(a)** $\theta_e = \theta_t = S_c = 0.1$

**(b)** $\theta_e = \theta_t = S_c = 0.8$

**Figure 2.** The choice of the government subsidy policies under the agency selling strategy.

1.  When the government subsidy level is low, the third-party platform's profits and the green R&D level reach the maximum value when the government subsidizes consumers' green consumption. If the service commission fee for selling products is small, the manufacturer can also obtain maximum profits when the government subsidizes consumers' green consumption. With an increase in the service commission fee for selling products, subsidizing the third-party platform's DDM by the government can bring higher profits to the manufacturer; however, the green R&D level is low at this time.
2.  When the government subsidy level is high, the third-party platform can obtain maximum profits when the government subsidizes consumers' green consumption. If the service commission fee of selling products is small, the manufacturer can obtain the largest profits when the government subsidizes the consumers' green

consumption, but the green R&D level is the highest when the government subsidizes the manufacturer's green R&D.

From Figure 3, we can verify the reselling strategy:

1.  When the government subsidy level is low, the profits of the manufacturer and the third-party platform can reach the maximum value when the government subsidizes consumers' green consumption, and the green R&D level can reach the maximum value when the government subsidizes the manufacturer's green R&D.

2.  When the government subsidy level is high but the sensitivity coefficient of consumers to green R&D activities is small, subsidizing consumers' green consumption by the government can help the supply chain members obtain the greatest economic benefits; meanwhile, the supply chain system can obtain more ecological benefits. With the increase in the sensitivity coefficient of consumers to green R&D activities, the platform supply chain members can obtain the greatest economic benefits; at the same time, the supply chain system can obtain more ecological benefits when the government subsidizes the manufacturer's green R&D.

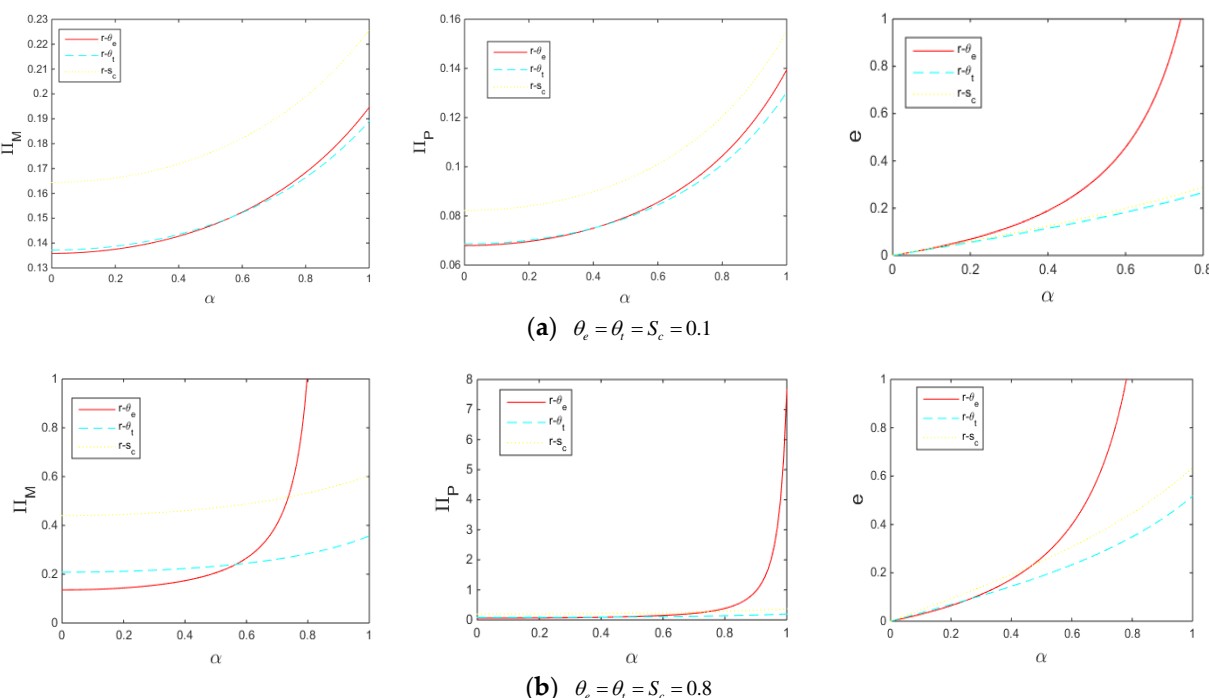

**Figure 3.** The choice of the government subsidy policies under the reselling strategy.

### 5.2. The Online Selling Strategy

#### 5.2.1. The Manufacturer's Preferred Selling Strategy

In the platform supply chain subsidized by the government, the manufacturer's preferred selling strategies are as follows.

From Figure 4 we can establish what happens when the unit service commission fee of selling products is small and the government subsidy level is low; the manufacturer should choose the agency-selling strategy of the government subsidizing consumers' green consumption. With the increase in government subsidy level, when the sensitivity coefficient of consumers to green R&D activities is large, the manufacturer chooses the agency-selling strategy of the government subsidizing the manufacturer's green R&D. When the sensitivity coefficient of consumers to DDM activities is great, the manufacturer chooses the reselling strategy of the government subsidizing the third-party platform's DDM. In other cases, the manufacturer prefers the agency-selling strategy of the government subsidizing consumers' green consumption.

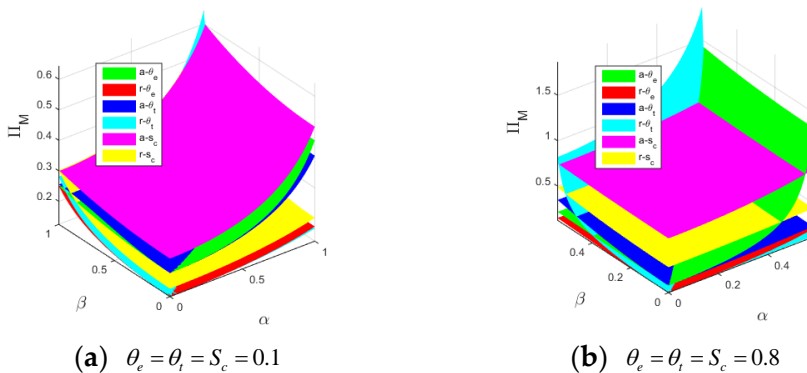

(**a**)  $\theta_e = \theta_t = S_c = 0.1$    (**b**)  $\theta_e = \theta_t = S_c = 0.8$

**Figure 4.** The influence of the government subsidy level on the manufacturer's choice of selling strategy when $u = 0.1$.

From Figure 5, we can see that when the unit service commission fee of selling products is moderate, if the government subsidy level is low and the sensitivity coefficient of consumers to DDM activities is small, the manufacturer chooses the reselling strategy of the government subsidizing consumers' green consumption. In other cases, the manufacturer chooses the agency-selling strategy of the government subsidizing consumers' green consumption. With the increase in the government subsidy level, if the sensitivity coefficient of consumers to DDM activities is large, the manufacturer chooses the agency-selling strategy of the government subsidizing the third-party platform's DDM. In other cases, the manufacturer prefers the agency-selling strategy of the government subsidizing consumers' green consumption.

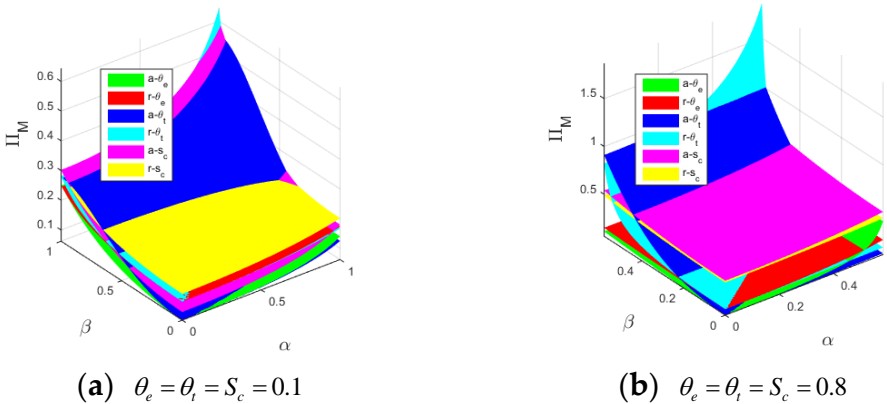

(**a**)  $\theta_e = \theta_t = S_c = 0.1$    (**b**)  $\theta_e = \theta_t = S_c = 0.8$

**Figure 5.** The influence of the government subsidy level on the manufacturer's choice of selling strategy, when $u = 0.5$.

From Figure 6, we can see that when the unit service commission fee for selling products is large, if the government subsidy level is low and the sensitivity coefficient of consumers to DDM activities is small, the manufacturer chooses the reselling strategy of the government subsidizing consumers' green consumption. In other cases, the manufacturer chooses the agency-selling strategy of the government subsidizing consumers' green consumption. With the increase in the government subsidy level, when the sensitivity coefficient of consumers to DDM activities is small, the manufacturer chooses the reselling strategy of the government subsidizing consumers' green consumption. When the sensitivity coefficient of consumers to DDM activities is large, the manufacturer prefers the reselling strategy of the government subsidizing the third-party platform's DDM.

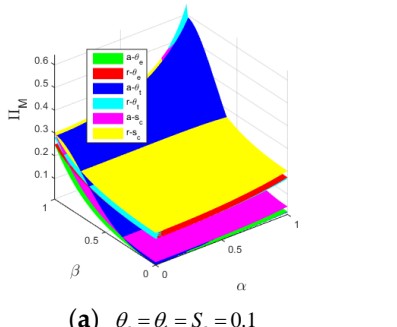
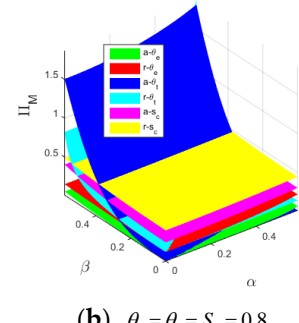

**(a)** $\theta_e = \theta_t = S_c = 0.1$    **(b)** $\theta_e = \theta_t = S_c = 0.8$

**Figure 6.** The influence of government subsidy level on the manufacturer's choice of selling strategy, when $u = 0.8$.

### 5.2.2. The Third-Party Platform's Preferred Selling Strategy

In the platform supply chain subsidized by the government, the third-party platform's preferred selling strategies are as follows.

From Figure 7, we can see that when the unit service commission fee for selling products is small, if the government subsidy level is low but the sensitivity coefficients of consumers to green R&D and DDM activities are both large, the third-party platform chooses the reselling strategy of the government subsidizing the third-party platform's DDM. In other cases, the third-party platform chooses the reselling strategy of the government subsidizing consumers' green consumption. With the increase in the government subsidy level, when the sensitivity coefficient of consumers to green R&D activities is large, the third-party platform chooses the reselling strategy of the government subsidizing the manufacturer's green R&D. When the sensitivity coefficient of consumers to DDM activities is large, the third-party platform selects the reselling strategy of the government subsidizing the third-party platform's DDM. In other cases, the third-party platform prefers the reselling strategy of the government subsidizing consumers' green consumption.

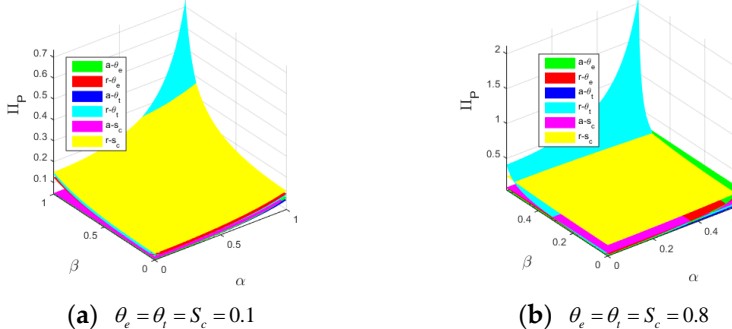

**(a)** $\theta_e = \theta_t = S_c = 0.1$    **(b)** $\theta_e = \theta_t = S_c = 0.8$

**Figure 7.** The influence of the government subsidy level on the third-party platform's choice of selling strategy, when $u = 0.1$.

From Figure 8, we can see that when the unit service commission fee for selling products is large, if the government subsidy level is low but the sensitivity coefficients of consumers to the green R&D and DDM activities are both large, the third-party platform prefers the reselling strategy of the government subsidizing the third-party platform's DDM. When the sensitivity coefficient of consumers to green R&D activities is small but the sensitivity coefficient of consumers to DDM activities is large, the third-party platform chooses the reselling strategy of the government subsidizing consumers' green consumption. In other cases, the third-party platform prefers the agency selling strategy of the government subsidizing consumers' green consumption. With the increase in the government subsidy level, when the sensitivity coefficient of consumers to green R&D activities is large, the third-party platform selects the reselling strategy of the government

subsidizing the third-party platform's DDM. In other cases, the third-party platform prefers the reselling strategy of the government subsidizing consumers' green consumption.

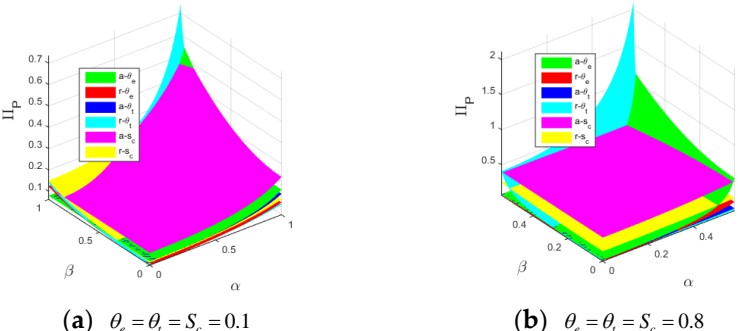

(**a**) $\theta_e = \theta_t = S_c = 0.1$      (**b**) $\theta_e = \theta_t = S_c = 0.8$

**Figure 8.** The influence of the government subsidy level on the third-party platform's choice of selling strategy, when $u \geq 0.4$.

### 5.2.3. The Equilibrium Selling Strategy

From Figure 9, we can see that when the unit service commission fee for selling products is small, if the government subsidy level is low but the sensitivity coefficients of consumers to green R&D and DDM activities are both large, the platform supply chain member can obtain greater economic benefits and the platform supply chain system can obtain more ecological benefits under the reselling strategy of the government subsidizing the third-party platform's DDM. The government prefers the strategy of subsidizing those supply chain members who can obtain the greatest ecological benefits; therefore, in other cases, the equilibrium strategy is the agency selling strategy of the government subsidizing the manufacturer's green R&D.

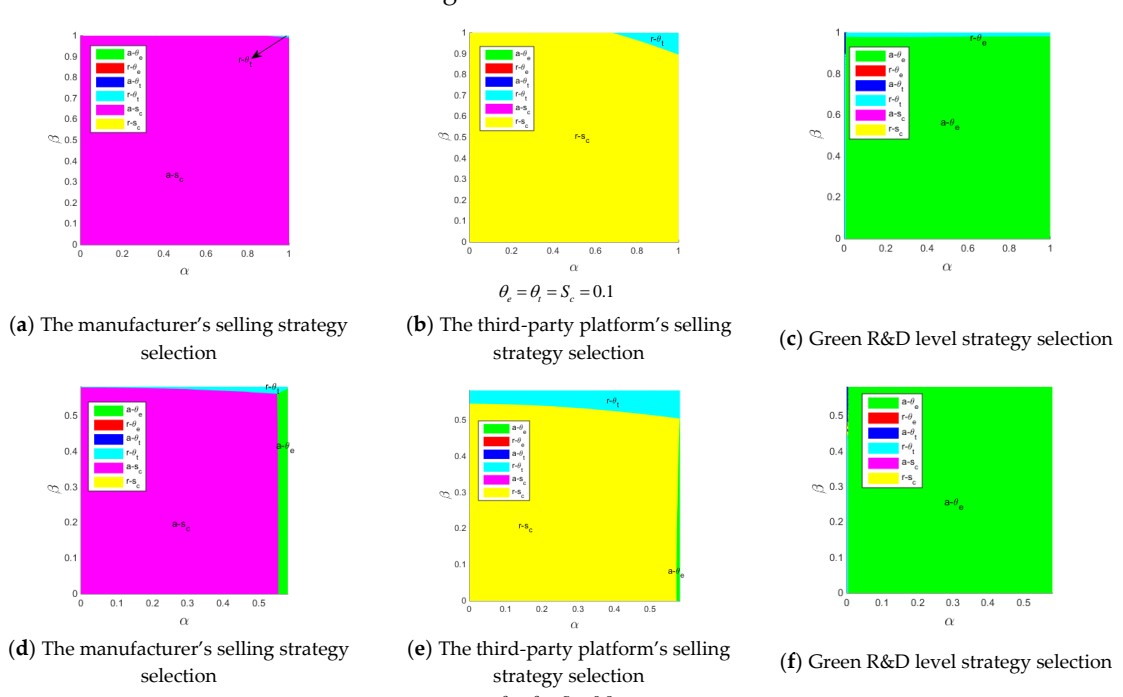

(**a**) The manufacturer's selling strategy selection

(**b**) The third-party platform's selling strategy selection

(**c**) Green R&D level strategy selection

$\theta_e = \theta_t = S_c = 0.1$

(**d**) The manufacturer's selling strategy selection

(**e**) The third-party platform's selling strategy selection

(**f**) Green R&D level strategy selection

$\theta_e = \theta_t = S_c = 0.8$

**Figure 9.** The equilibrium strategy of the supply chain, when $u = 0.1$.

From Figure 10 we can see that when the unit service commission fee for selling products is moderate, if the government subsidy level is small and the sensitivity coefficients of consumers to green R&D and DDM activities are both small, under the agency-selling strategy of subsidizing consumers' green consumption, the platform supply system can obtain the greatest ecological benefits. At this point, the third-party platform can obtain

more economic benefits. In other cases, the equilibrium strategy is the agency-selling strategy of subsidizing the manufacturer's green R&D because the platform supply chain system can obtain the greatest ecological benefits under this strategy:

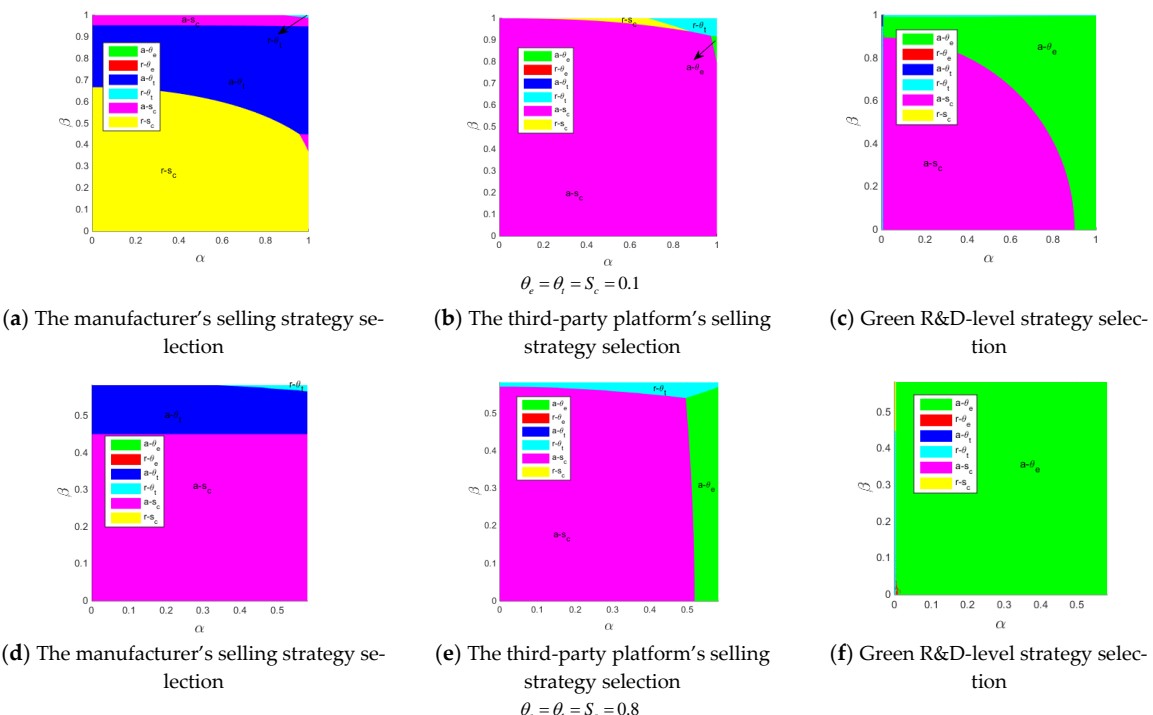

**Figure 10.** The equilibrium strategy of the supply chain when $u = 0.5$.

From Figure 11, we can see that if the sensitivity coefficients of consumers to green R&D and DDM activities are both small, the platform supply chain system can obtain the greatest ecological benefits under the reselling strategy of the government subsidizing the manufacturer's green R&D. Otherwise, the equilibrium strategy is the agency-selling strategy of the government subsidizing the manufacturer's green R&D.

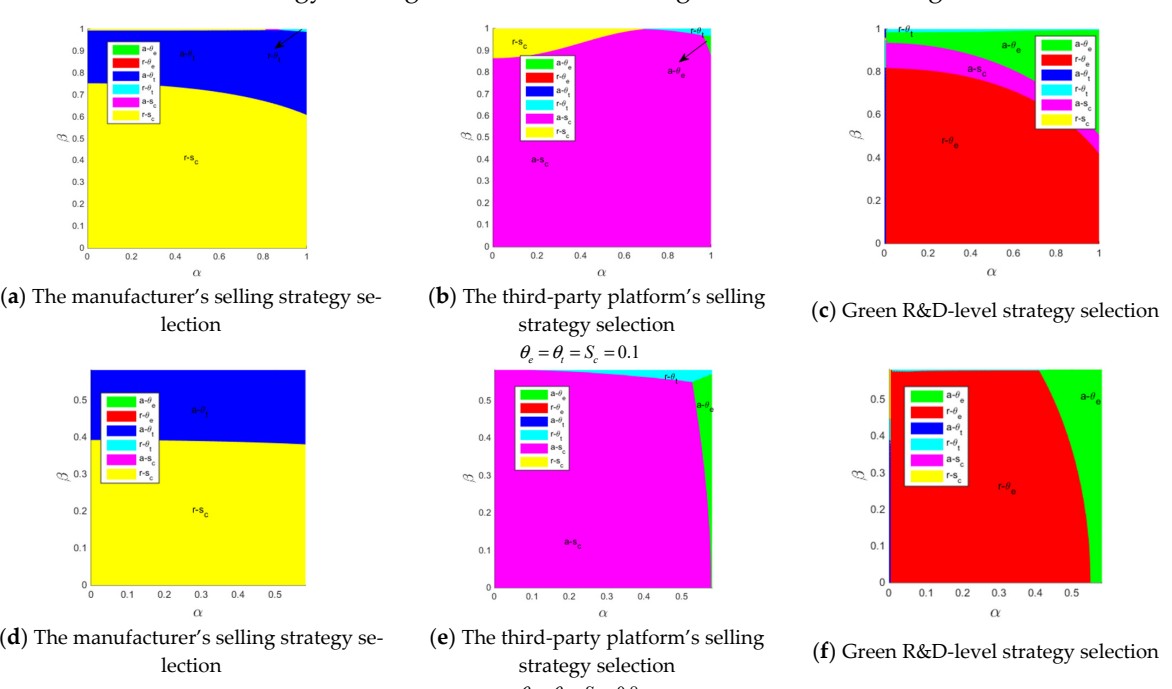

**Figure 11.** The equilibrium strategy of the supply chain, when $u = 0.8$.

## 6. Conclusions

In the platform supply chain with the agency selling or reselling services, we constructed six models including government subsidizing the manufacturer's green R&D, the third-party platform's DDM, and consumers' green consumption. Then, we compared and analyzed the results of three government subsidy policies by modeling, game theory, and numerical analysis, identifying which government subsidy policy and online selling strategy could improve the supply chain members' economic benefits and the supply chain system's ecological benefits.

The following points are obtained from an analysis of the research in this paper:

1.  As the sensitivity coefficient of consumers to green R&D activities increases, consumers are more willing to buy green products, which will increase the green products' prices and encourage the manufacturers to carry out green R&D activities to produce green products. At this point, the manufacturer's profits and the green R&D level are increasing. As a survey of Chinese consumers' green consumption awareness shows, the willingness of consumers to buy green products rose from 58% to 83%, which undoubtedly stimulated manufacturers to carry out green R&D activities and produce more green products. As the sensitivity coefficient of consumers to the DDM activities increases, the third-party platform is encouraged to improve the DDM level. Therefore, as the unit selling price of green products increases, the platform charges a higher service commission fee for selling products; finally, the third-party platform's profit increases. That is, the more sensitive consumers are to green R&D and DDM activities, the more platform supply chain members can obtain economic and ecological benefits.

2.  Different government subsidy policies have different impacts on the third-party platform's economic and ecological benefits: (1) under the agency-selling strategy, when the government subsidy level is low and the service commission fee for selling products is small, the profits of the manufacturer and the third-party platform, and the green R&D level all reach the maximum value when the government subsidizes consumers' green consumption. That is, the subsidizing of consumers' green consumption by the government can not only help the supply chain members to obtain greater economic benefits but also help the platform supply chain system to improve the ecological benefits. With an increase in the service commission fee for selling products, the manufacturer can achieve higher profits when the government subsidizes the third-party platform's DDM, but great ecological benefits cannot be obtained at this time. When the government subsidy level is high, the subsidizing of the consumers' green consumption by the government can bring higher profits to the platform supply chain members. If the service commission fee for selling products is small, the government can also bring higher profits to the manufacturers by subsidizing consumers' green consumption. This ecological benefit cans reach the maximum value when the government subsidizes the manufacturer's green R&D. (2) Under the reselling strategy, when the government subsidy level is low, the manufacturer and the third-party platform can obtain the maximum profits when the government subsidizes consumers' green consumption, while the green R&D level is the highest when the government subsidizes the manufacturer's green R&D. That is, the platform supply chain system can obtain the maximum economic benefits but achieve fewer ecological benefits. When the level of government subsidy is high, if the sensitivity coefficient of consumers to green R&D is small, the profits of the manufacturer and the tiered-party platform and of the green R&D level are the largest when the government subsidizes consumers' green consumption. In this case, the platform supply chain system can achieve win-win economic and ecological benefits. Just as the Chinese government has issued a large number of high-value vouchers to consumers who purchase Haier's energy-saving products on JD.COM, this stimulates consumers to buy more green products and encourages the manufacturer to produce more green products. Moreover, JD.COM adopts a reselling strategy for selling Haier's products,

which is consistent with our conclusion. As the sensitivity coefficient of consumers to green R&D activities increases, the profits of the manufacturer and the third-party platform and the green R&D level can all reach the maximum value when the government subsidizes the manufacturer's green R&D. In other words, when the government subsidizes the manufacturer's green R&D, the platform supply chain can achieve win-win economic and ecological benefits. In fact, as consumers become more sensitive to green R&D activities, the German government has attracted EUR 14 billion to the manufacturer to produce green semiconductor chips, and these green chips are resold on Amazon.com using the platform's DDM advantages.

3. Through the online selling strategy analysis of the platform supply chain, we can see that: (1) for the manufacturer, when the unit service commission fee for selling products is small, the manufacturer is more willing to choose the agency-selling strategy of the government subsidizing consumers' green consumption, otherwise the manufacturer prefers the agency-selling strategy of the government subsidizing the third-party platform's DDM. With the increase in the unit service commission fee for selling products, the manufacturer gradually chooses the reselling strategy of the government subsidizing the manufacturer's green R&D. If the sensitivity coefficient of consumers to green R&D activities is large, the manufacturer chooses the agency selling strategy of government subsidizing the third-party platform's DDM. (2) For the third-party platform, when the unit service commission fee of selling products is small, the third-party platform is more willing to choose the agency-selling strategy of the government subsidizing consumers' green consumption. With the increase in the unit service commission fee for selling products, the third-party platform prefers to choose the reselling strategy of the government subsidizing consumers' green consumption. When the sensitivity coefficient of consumers to green R&D activities is large, the third-party platform is more willing to choose the agency-selling strategy of the government subsidizing the manufacturer's green R&D. If the sensitivity coefficient of consumers to DDM activities is large, the third-party platform chooses the reselling strategy of the government subsidizing the third-party platform's DDM. (3) For the platform supply chain system, subsidizing the manufacturer's green R&D by the government is most likely to obtain the greatest ecological benefits. When the unit service commission fee of selling products is small, the third-party platform's provision of agency selling services can better improve the ecological benefits. When the unit service commission fee of selling products is large, the third-party platform's provision of reselling services can achieve the same effect. Therefore, the different service commission fees for selling products can lead the manufacturer and the third-party platform to choose different selling strategies. The government prefers the strategy of subsidizing the supply chain members who can obtain a higher green R&D level. On this basis, the third-party platform can further help the supply chain system to improve ecological benefits by providing different selling services.

This paper can further elucidate the following aspects. First, supply chains with multiple members have become very common; therefore, on the basis of our paper, future research can consider two or more competing manufacturers and platforms. Second, with the rapid development of the platform supply chain, providing both agency-selling and reselling services for the manufacturers is more realistic. Then, in addition to a subsidy strategy where the government directly subsidizes the supply chain members, we can further incorporate other government subsidy strategies, such as punitive measures and taxation measures into the research model for analysis. Besides, the platform's DDM advantages can help the manufacturer to reduce operating costs while increasing product sales, which is a very interesting possibility to explore.

**Author Contributions:** Conceptualization, Z.M., G.Z; methodology, Z.M.; software, Q.L., F.Z.; validation, Q.L.; formal analysis, Z.M., G.D.; investigation, Q.L., K.L.; resources, Q.L., F.Z.; data curation, Z.M.; writing—original draft preparation, Q.L.; writing—review and editing, Q.L.; visualization, Z.M., G.D.; supervision, Q.L.; project administration, Q.L., G.D.; funding acquisition, Z.M., G.Z. All authors have read and agreed to the published version of the manuscript.

**Funding:** This work was supported by the Outstanding Youth Innovation Team Project of Colleges and Universities in Shandong Province under Grant Nos. 2020RWG011.

**Institutional Review Board Statement:** Not applicable.

**Informed Consent Statement:** Not applicable.

**Data Availability Statement:** Not applicable.

**Conflicts of Interest:** The authors declare no conflict of interest.

## Appendix A

The proof of the equilibrium decisions of the platform supply chain under the agency-selling strategy of the government subsidizing the manufacturer's green R&D in Proposition 1 is as follows.

Through backward induction, we first take the derivative of the third-party platform's profit function with respect to $t$ and set it zero; that is, $\frac{\partial \Pi_P^{a-\theta_e}(t)}{\partial t} = 0$; we further obtain the second-order condition $\frac{\partial^2 \Pi_P^{a-\theta_e}(t)}{\partial t^2} = -1 < 0$. Therefore, the third-party platform's profit is a concave function of $t$, and the solution is $t^{a-\theta_e} = \beta u$. By substituting the $t^{a-\theta_e}$ into the manufacturer's profit and taking the derivative with respect to $p$ and $e$, then we can establish:

$$\frac{\partial \Pi_M^{a-\theta_e}(p,e)}{\partial p} = \beta^2 u + u + \alpha e - 2p + 1 \tag{A1}$$

$$\frac{\partial \Pi_M^{a-\theta_e}(p,e)}{\partial e} = (p-u)\alpha - (1-\theta_e)e \tag{A2}$$

Then we set Equations (A1) and (A2) to 0; that is, $\frac{\partial \Pi_M^{a-\theta_e}(p,e)}{\partial p} = 0$ and $\frac{\partial \Pi_M^{a-\theta_e}(p,e)}{\partial e} = 0$, where we can establish $\begin{cases} \beta^2 u + u + \alpha e - 2p + 1 = 0 \\ (p-u)\alpha - (1-\theta_e)e = 0 \end{cases}$. We further obtained the second-order condition:

$$\frac{\partial^2 \Pi_M^{a-\theta_e}(p,e)}{\partial p^2} = -2 \tag{A3}$$

$$\frac{\partial^2 \Pi_M^{a-\theta_e}(p,e)}{\partial p \partial e} = \alpha \tag{A4}$$

$$\frac{\partial^2 \Pi_M^{a-\theta_e}(p,e)}{\partial e \partial p} = \alpha \tag{A5}$$

$$\frac{\partial^2 \Pi_M^{a-\theta_e}(p,e)}{\partial e^2} = -(1-\theta_e) \tag{A6}$$

Through Equations (A3)–(A6), we can establish the Hessian matrix $\Pi_M^{a-\theta_e}(p,e)$ is $\begin{pmatrix} -2 & \alpha \\ \alpha & -(1-\theta_e) \end{pmatrix}$. Therefore, if $\theta_e < 1 - \frac{\alpha^2}{2}$, the profit function of the manufacturer's profit is concave about $p$ and $e$, so the constraints need to be satisfied throughout our paper.

By these simultaneous equations $\begin{cases} \beta^2 u + u + \alpha e - 2p + 1 = 0 \\ (p - u)\alpha - (1 - \theta_e)e = 0 \end{cases}$ we can obtain:

$$\begin{cases} p^{a-\theta_e *} = \frac{\left[(1+\beta^2)(1-\theta_e)-\alpha^2\right]u+(1-\theta_e)}{2(1-\theta_e)-\alpha^2} \\ e^{a-\theta_e *} = \frac{\alpha\left[1-u\left(1-\beta^2\right)\right]}{2(1-\theta_e)-\alpha^2} \end{cases}$$ . By substituting $p^{a-\theta_e *}$ and $e^{a-\theta_e *}$ into $t^{a-\theta_e}$, $D^{a-\theta_e}$,

$\Pi_M^{a-\theta_e}$, $\Pi_P^{a-\theta_e}$, Proposition 1 can be confirmed.

Under the reselling mode, the third-party platform sells green products to consumers at a marked-up price for green products; that is, the profit function $\Pi_P^{r-\theta_e}(p,t)$ is equal to the profit function $\Pi_P^{r-\theta_e}(m,t)$.

Propositions 2–6 can be obtained by a similar proof process, so we omit them.

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
