# Peer review of "Government Subsidy Policy and Online Selling Strategy in a Platform Supply Chain with Green R&D and DDM Activities"

_sustainability, doi:10.3390/su14159658_

Round 1

Reviewer 1 Report

I appreciate your efforts and the work put into the data collection and analysis. I think you made a great decision to provide the premises and the results of your study in the first part of the paper (Line 89-100). Likewise, the study is well written and the structure of the article is transparently presented. However, I would like to suggest some changes and improvements.

First of all, the abstract is the first section of your research. Accordingly, it should offer a more comprehensive and accurate summarize of the content. Therefore, I recommend including here also the methodology used in the study and the arguments for the method chosen.

In particular, the paper presents briefly in the introduction part some specific cases from around the world: the German and Chinese governments strategies in terms of encouraging a more environmentally friendly production (Line 33), the Swedish government subsidy policy for green cars (Line 37). Although some of the aspects have been pointed, I was expected to read a more theory-oriented knowledge on the negative impact of manufacturing activities on carbon emission and on government intervention. I would strongly suggest that you must go beyond short and descriptive examples of some case at an international level and embody a more critical and analytical approach in relation to the area of concern – green R&D operations in a direct relation with data-drive marketing  activities and the influence upon carbon emissions.

On the positive side, the literature review is consistent enough, the authors not confining themselves just to summarize some references, but discussing them critically. As a result, the author evaluated both studies from the green manufacturing topic: Chen et al. (2008) – Line 114, Hong et al. (2019) – Line 129, Yang et al. (2017) – Line 159, as well as the platform supply chain management: Li (2020) – Line 196, Wang et al. (2016) - Line 199, Braverman (2015) – Line 226. Be that as it may, I suggest summarizing and highlighting the strengths and weaknesses of prior work in the topic of green research & development manufacturing process and how it can negatively affect the environment and the carbon emissions.

Furthermore, the methodology used in the paper is explained properly. For instance, lines 260-319 presents in detail six manufacturing supply chain models, along with the selling strategies and the third-party platforms involved. In addition, Figure 1 (Line 287) presents a visualization map for each different model framework.

Equally important, the numerical analysis is clearly presented, involving numerous graphic representations (Lines 485-648). Finally, the conclusions of the study are explicitly presented and are tied with the results. Specifically, the author included here the conclusions and arguments relating to every result obtained. Altogether, in terms of grammar the phrases used in the article are straightforward and easy to understand.

All things considered, authors need to present the limitations and future research directions of the study.

Reviewer 2 Report

These results claimed in this study will need to be verified through empirical analysis later. Thank you. 

Reviewer 3 Report

The paper is important and relevant to Sustainability. Following are my comments to improve the document: 

1. The problem statement should be improved. 

2. The originality of this study is not clear. 

3. The recommendations for practising managers are required with proper justification. 

Best of luck!
